# Acute respiratory infection rates in primary care anticipate ICU bed occupancy during COVID-19 waves

**Montserrat Guillen**[1]*, **Ignasi Bardes Robles**[2¤], **Ester Bordera Cabrera**[2¤], **Xénia Acebes Roldán**[2¤], **Catalina Bolancé**[1], **Daniel Jorba**[3], **David Moriña**[4]

**1** Riskcenter-IREA, Department of Econometrics, University of Barcelona, Barcelona, Spain, **2** CatSalut Care Management Direction, Generalitat de Catalunya, Barcelona, Spain, **3** Faculty of Medicine and Health Sciences, University of Barcelona, Barcelona, Spain, **4** Department of Econometrics, University of Barcelona, Barcelona, Spain

¤ Current address: Hospital de la Santa Creu i Sant Pau, Barcelona, Spain
* mguillen@ub.edu

## Abstract

### Background

Bed occupancy in the ICU is a major constraint to in-patient care during COVID-19 pandemic. Diagnoses of acute respiratory infection (ARI) by general practitioners have not previously been investigated as an early warning indicator of ICU occupancy.

### Methods

A population-based central health care system registry in the autonomous community of Catalonia, Spain, was used to analyze all diagnoses of ARI related to COVID-19 established by general practitioners and the number of occupied ICU beds in all hospitals from Catalonia between March 26, 2020 and January 20, 2021. The primary outcome was the cross-correlation between the series of COVID-19-related ARI cases and ICU bed occupancy taking into account the effect of bank holidays and weekends. Recalculations were later implemented until March 27, 2022.

### Findings

Weekly average incidence of ARI diagnoses increased from 252.7 per 100,000 in August, 2020 to 496.5 in October, 2020 (294.2 in November, 2020), while the average number of ICU beds occupied by COVID-19-infected patients rose from 1.7 per 100,000 to 3.5 in the same period (6.9 in November, 2020). The incidence of ARI detected in the primary care setting anticipated hospital occupancy of ICUs, with a maximum correlation of 17.3 days in advance (95% confidence interval 15.9 to 18.9).

### Interpretation

COVID-19-related ARI cases may be a novel warning sign of ICU occupancy with a delay of over two weeks, a latency window period for establishing restrictions on social contacts and

**Data Availability Statement:** Data are available from the public institutional repository at http://www.ub.edu/rfa/R/covid Data are also available at

http://dx.doi.org/10.17504/protocols.io.
36wgq7de5vk5/v1.

**Funding:** This study was supported by grants from the Spanish Ministry of Science and Innovation (code PID2019-105986GB-C21) and Fundacion BBVA (MG, CB) and Instituto de Salud Carlos III (code COV20/00115) (DM), Madrid, Spain, and Institució Catalana de Recerca i Estudis Avançats ICREA Academia (MG), Barcelona, Spain". The funders had no role in study design, data collection and analysis, decision to publish, or preparation of the manuscript.

**Competing interests:** The authors have declared that no competing interests exist.

mobility to mitigate the propagation of COVID-19. Monitoring ARI cases would enable immediate adoption of measures to prevent ICU saturation in future waves.

## Introduction

Saturation of hospital and especially ICU beds is the reason to declare nationwide lockdowns during COVID-19 pandemic [1, 2]. Early warnings to predict hospital occupancy waves mostly rely on the number of confirmed cases, contagion rates, and adherence to self-isolation measures [3–7]. These figures, however, are affected by changes in social behavior, screening strategies, and testing methodologies that are not necessarily stable over the course of the pandemic [8, 9]. Many reports have shown that prompt interventions can prevent the surge in the number of cases [10–14], but early timing of the imposition of restrictions may have devastating economic consequences. Therefore, there is an urgent need to identify fast, consistent, and reliable indicators for anticipating the resurge [15] or decline [16] of a new wave of COVID-19. Despite huge research efforts around the world to fight against COVID-19 pandemic, the role of general practitioners has not been extensively investigated, although they are on the front line for identifying patients with COVID-19 presenting with mild symptoms.

Since March 26, 2020, general practitioners from the primary health care setting in Catalonia, Spain, were enforced to report all cases of acute respiratory infections caused by COVID-19, flu, or other causes. The objective of this study was to assess whether the series of daily cases of COVID-19-related diagnoses of ARI could anticipate the behavior of the series of ICU bed occupancy, so that the former figure could be an early warning indicator of potential inpatient care saturation.

## Methods

### Design and setting

This was a retrospective study of data recorded prospectively in the registry of the Integrated Public Health System of Catalonia (SISCAT), which provides universal free full-health care coverage for the citizens of Catalonia. Catalonia is an autonomous community on the northeastern part of Spain (official 2020 population, 7,727,029 inhabitants) and a geographical area of 32,108 km$^2$ about the size of Belgium. The majority of the population lives in Barcelona (the capital) and its metropolitan area and along the Mediterranean coast, while counties in the interior are much less densely populated (overall population density 241/km$^2$). There are 434 primary health care centers in the public network, with health care provided by general practitioners who are specialists in family and community medicine, pediatricians, nurses, and social workers. These centers, also linked to the network of public county and regional hospitals, are usually closed during weekends and bank holidays, but there are emergency centers in the primary care setting providing 24/7 service.

For the purpose of this registry-based study, the requirement of a written informed consent was waived. The researchers only analyzed anonymized and aggregated data. The Clinical Research Ethics Committee of Hospital Universitari de Bellvitge in L'Hospitalet de Llobregat, Barcelona (Spain) approved the study (date December 31, 2020, reference PR454/20).

### Participants and definitions

All general practitioners from the primary care network were required to register cases of their patients diagnosed with ARI. ARI was defined as an infection that may interfere with normal

breathing. It can affect just the upper respiratory system, or just the lower respiratory system (ICD10 codes related to SNOMED CT: influenza, adenoviral respiratory disease, viral respiratory infection, viral upper respiratory tract infection, viral lower respiratory infection, viral pleurisy, and severe acute respiratory syndrome; flu terms were excluded). The diagnosis of ARI was made clinically by the general practitioner. The infection was assumed to be related to COVID-19 only if a confirmed positive result was obtained by a real-time reverse-transcriptase-polymerase chain reaction (rRT-PCR) and/or antigen testing. General practitioners were also required to register the date when ARI was suspected and confirmed to be related to COVID-19, flu, or other causes. Once patients were reported, they were not registered again if they went to see their general practitioner for subsequent visits.

The study data form recording COVID-19-related ARI cases was completed by the physicians in charge of the patients, and the data were stored in the registry system on the SISCAT database server. Consistency of data was confirmed by SISCAT.

## Data collection

Data were collected between March 25, 2020 and January 20, 2021, and included aggregated total cases of COVID-19-related ARI diagnosed in the primary care setting and registered in the SISCAT database and total number of ICU beds occupied by COVID-19 patients. The number of cases and the number of beds were calculated per 100,000 residents in order to provide results comparable to other regions. To evaluate the stability of the results, data collection was extended to March 27, 2022.

The total number of cases of ARI related to COVID-19 was gathered daily and the time series was updated according to SARS–CoV-2 test results. Daily series of occupied ICU beds in all Catalonian hospitals (both private and public) was also compiled for this study. At midnight, information regarding the number of beds that were occupied in each hospital was sent to the central system as well as the daily total of ICU beds, from which the time series of ICU occupancy was compiled. The series of total COVID-19 detected cases was not evaluated because the strategies and intensity of screening varied throughout the observation period. For example, patients with mild symptoms or even those that are asymptomatic (approximately one third) [17] did not always visit a general practitioner.

The total number of ICU beds available in Catalonia is over 11.6 per 100,000 inhabitants (around 900). However, in Spring 2020, hospitals had to convert hospital areas into ICUs due to a dramatic increase in COVID-19-infected patients, as a result of which the total ICU occupancy related to COVID-19 reached as much as 19.7 per 100,000 inhabitants (1,528 beds) [18].

## Outcome measures

The primary outcome measure was the cross-correlation between the series of COVID-19-related ARI cases and ICU bed occupancy by COVID-19-infected patients, taking into account the effect of bank holidays and weekends, expressed as the delay (in days) from the point of maximum correlation between the series of ARI cases per 100,000 inhabitants and ICU bed occupancy per 100,000 inhabitants.

## Statistical analysis

The daily series of ARI rates and the daily series of ICU occupancy rates were compared visually. The series of ARI showed weekly peaks that corresponded to a high value of cases occurring on Mondays (or first day after a Monday bank holiday) and a low value corresponding to weekends and holidays. The series of ICU occupancy was smooth as it reflected the inflow of new admissions and discharges. Data from April 1, 2020, onwards were selected to eliminate the lack of

complete ICU data of March 2020, and to be able to analyze up to a one month gap between ARI and ICU peaks. Firstly, the series of ARI cases was filtered to remove the effect of the first working day of the week, holidays, and weekends via a multivariate log-linear regression model with the corresponding binary variables. The lagged ARI series and the filtered series were compared with the series of ICU occupancy. The lags from 1 to up to 30 days were analyzed to compare the dynamics of ARI with those showing up later in ICU occupancy. The series of ICU occupancy rates and the series of unfiltered (and filtered) ARI rates were analyzed. The optimal lag was found by maximizing the linear cross-correlation between the ICU and the ARI series. By interpolating the results, an optimal fractional lag was obtained. Correlation was assessed with the Pearson's correlation coefficient ($r$). Statistical significance was set at $P < 0.05$.

To provide a confidence interval, a bootstrap method was used. Each replicated series of ARI and ICU was the result of a bootstrapped version of the time series using the Box-Cox and Loess-based decomposition bootstrap [19]. The analysis was repeated and the empirical 95% highest and 5% lowest maximum correlation lag values were provided as bootstrap confidence interval estimates. All statistical analyses were performed with the use of the R program, version 3.6.1. All tests were two-tailed, and the number of replicates in bootstrap confidence interval calculation was 2,000.

## Results

### Time series of daily indicators

During the study period, a delay appeared between ARI case waves and ICU waves (Fig 1). After the first wave, the weekly incidence of ARI cases increased from 252.7 per 100,000 in August 2020 to 496.5 in October 2020 (294.2 in November 2020), while the average number of ICU beds occupied by COVID-19 patients rose from 1.7 per 100,000 to 3.5 in the same period and jumped to 6.9 per 100,000 in November 2020 (Table 1).

### Outcomes

For the unfiltered series of COVID-19-related ARI rates and the series of ICU rates, the maximum cross-correlation was obtained at lag 18 ($r = 0.40$, $P < 0.001$). Interpolation indicated

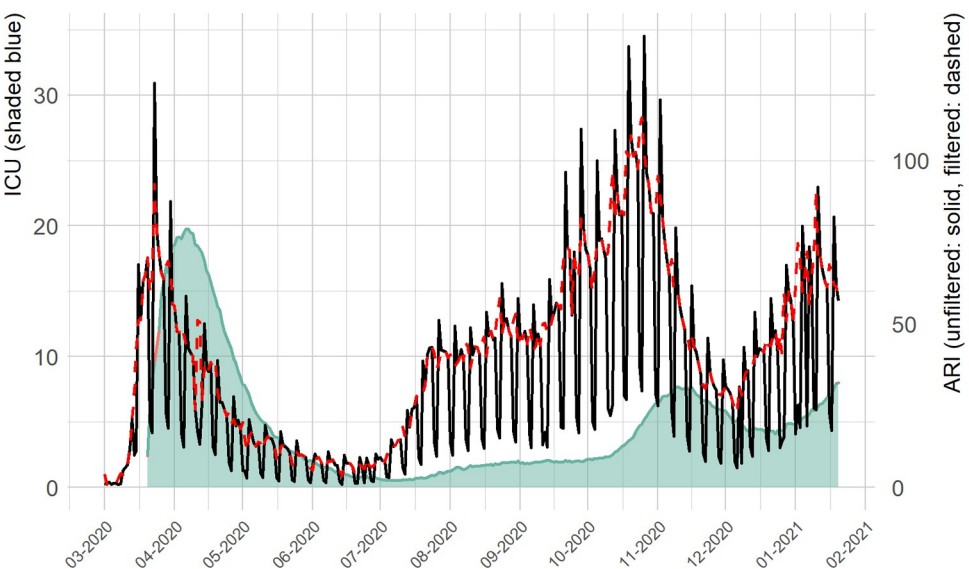

**Fig 1. Daily series of acute respiratory infection rates (cases per 100,000 inhabitants, solid line), filtered series (dashed line) and ICU occupancy (beds per 100,000 inhabitants, shaded) by date (mm-yyyy).**

**Table 1. Average weekly incidence of Acute Respiratory Infection (ARI) cases with Codid-19 diagnosed in Primary Care and Average Intensive Care Unit (ICU) Bed Occupancy of Covid-19 patients, all per 100,000 habitants by months in Catalonia, Spain.**

| Month (year 2020) | ARI related to Covid-19 diagnosed in Primary Care per 100,000 (average weekly incidence) | ICU Covid-19 beds per 100,000 (average occupancy) |
|---|---|---|
| April | 200.7 | 15.5 |
| May | 69.2 | 4.2 |
| June | 43.6 | 1.1 |
| July | 153.7 | 0.7 |
| August | 252.7 | 1.7 |
| September | 333.3 | 2.0 |
| October | 496.5 | 3.5 |
| November | 294.2 | 6.9 |
| December | 230.2 | 4.7 |

that the higher correlation was at 17.3 days (95% confidence interval [CI], 15.9 to 18.9). After filtering the effect of bank holidays and weekends in the ARI series, the highest lagged correlation ($r = 0.52$, $P < 0.001$) was found at 17.1 days, (95% CI, 16.7 to 17.3) which was over 2 weeks. Fig 2 shows the cross-correlation between the filtered ARI rates series and the ICU rates series. Table 2 extends the data timeframe. We present estimates of anticipation lag time in days and the corresponding CI, using a monthly timeframe rolling window until March 2022.

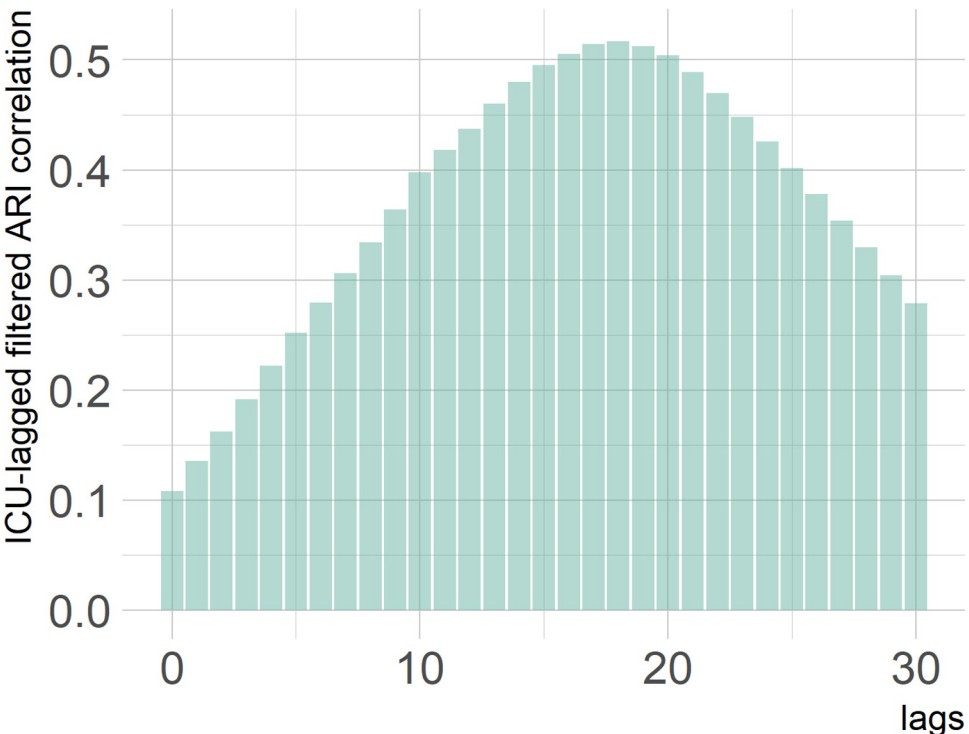

**Fig 2. Cross-correlogram between filtered ARI rates and ICU occupancy rates (right).** The x-axis indicates the number of lags.

**Table 2. Anticipation lag (in days) of Acute Respiratory Infection (ARI) cases with Codid-19 diagnosed in Primary Care (original and filtered) for Intensive Care Unit (ICU) Bed Occupancy of Covid-19 patients, in a rolling window from April 2020 to March 2022 in Catalonia, Spain (with confidence intervals, CI).**

| Data window | Anticipation lag of original ARI series (CI) | Anticipation lag of filtered ARI series (CI) |
|---|---|---|
| Apr. 20 –Jan. 21 | 17,3 (15,5;17,9) | 17,1 (16,5;17,7) |
| May 20- Feb. 21 | 18,9 (16,9;19,1) | 18,8 (18,6;19,1) |
| Jun. 20 –Mar.21 | 18,7 (18,5;19,1) | 18,8 (18,8;19,0) |
| Jul. 20 –Apr. 21 | 18,7 (17,2;19,1) | 18,8 (18,7;19,0) |
| Aug. 20- May 21 | 18,8 (16,8;19,1) | 19,0 (18,5;19,1) |
| Sep. 20 –Jun-21 | 18,3 (15,5;19,0) | 18,7 (18,6;19,2) |
| Oct. 20 –Jul. 21 | 18,7 (17,0;19,0) | 18,9 (18,6;19,0) |
| Nov 20 –Aug. 21 | 18,7 (17,2;19,2) | 18,6 (18,6;19,6) |
| Dec. 20 –Sep. 21 | 19,0 (17,2;19,2) | 18,6 (18,6;19,6) |
| Jan. 21- Oct. 21 | 19,0 (17,2;19,2) | 18,7 (18,6;19,6) |
| Feb. 21 –Nov. 21 | 18,8 (17,2;19,1) | 18,7 (18,6;19,0) |
| Mar. 21 –Dec. 21* | 19,0 (17,2;19,1) | 18,7 (18,6;19,5) |
| Apr. 21 –Jan. 22* | 18,7 (14,8;18,9) | 16,7 (17,1;18,9) |
| May 21 –Feb. 22* | 12,7 (11,9;12,7) | 12,6 (12,5;12,7) |
| Jun. 21 –Mar. 22* | 12,7 (11,6;12,7) | 12,7 (11,9;12,7) |

*Omicron.

## Discussion

According to the analysis of data from a regional population-based registry, this study found that ARI diagnoses related to COVID-19 infection made by general practitioners correlated significantly with ICU bed occupancy by COVID-19-infected patients over COVID-19 waves recorded in Catalonia in 2020. Interestingly, an increase in the number of ARI cases could anticipate an increase in ICU bed occupancy, with a delay of 17.3 days (slightly more than 2 weeks). This means to have available a consistent time gap for anticipating ICU occupancy saturation when a COVID-19 wave of cases appears to start. Therefore, there is sufficient time to impose restrictions to diminish social contacts and reduce the speed of contagion and the incidence of COVID-19-related ARI cases with subsequent decrease of COVID-19-related ICU bed occupancy.

We also found that ARI series is a simple and easy early warning signal, with the advantage that ARI series are gathered directly by general practitioners and accordingly does not depend upon the number of detected cases in the general population. Total cases might be influenced by the type of testing and screening strategies implemented in the region. For example, if most new cases affect a younger segment of the population, the incidence of COVID-19-related ARI and subsequent hospitalizations may not increase as much as the incidence of COVID-19 [20]. Underreporting of cases [21] and false-negative and false-positive COVID-19 test results [22, 23] discourage the use of series of total positive cases.

To our knowledge, only a few studies have assessed early signals associated with ICU requirements and some have only investigated Internet searches and social media data [24, 25]. A retrospective quantitative analysis from the Ile de France region in France (12.1 million inhabitants, population density equal 1,000/km$^2$) from the first wave (data from February 20 to May 5, 2020) [26] portrayed a few early indicators of the number of COVID19 patients requiring ICU care during the epidemic crisis, none of them being the identification of ARI cases. The only signal that was found to flag earlier than the series of daily ARI cases was the

daily number of COVID-19-related telephone calls received by the emergence medical services (EMS), showing a 23-day delay in the correlation curve. Dispatching ambulances, proportion of positive rRT-PCR tests, emergency department visits, and general practitioner visits were associated with COVID-19 ICU patients with an anticipation delay of 15, 14, 13, and 12 days, respectively. Qualitative analysis was reported to provide similar conclusions from August 1, 2020 to September 15, 2020, but no additional estimates were provided [26]. In relation to general practitioner visits, it should be noted that data corresponded to *SOS médicins*, that is, the number of COVID-19 diagnoses made by a private network of general practitioners who performed only emergency visits on a 24 hour and 7 day basis at home. This is a remarkably different perspective that ARI diagnoses made by general practitioners in the primary care setting of entire Catalonia. Additionally, the merits of the daily number of emergency calls received by the EMS needs to be confirmed for subsequent and/or smaller waves and its application to other countries and comparability could be challenged by the type of medical assessment provided and how the use of emergency resources is incited in different places. Moreover, EMS calls usually do not distinguish call-backs, call types such as low acuity or other than medical demands, and implementation of computerized triage [27]. EMS call services may be severely disrupted by peak number of calls and even blackouts [28], which suggests that most EMS are likely to need pandemic crisis redefinitions [27].

We performed a data analysis of the daily curves for a period longer than that reported in previous studies, in a less densely populated region and including the initial days of the third wave. It did not contain subregional or patient specific characteristics analysis. We assumed the full capacity to perform ARI diagnosis by general practitioners, the consistency of diagnostic criteria over the observation period, and the reliability of the SISCAT database.

This method has been successfully implemented by the local authorities. When Primary Care diagnoses of ARI increased (or decreased), subsequent ICU occupancy increase (or decreased) followed about three weeks later. This neat indicator did not depend on recorded infection rates (which could be contingent on test availability) and was only challenged when Primary Care facilities became fully saturated during the Omicron variant wave from December 2021 to January 2022. We expect that in future outbreaks similar analyses will help anticipating ICU occupancy.

With Omicron our analysis reveals a shorter anticipation lag in days, which could be explained by the fact that Primary Care facilities were overloaded at that time. We have considered the availability of outpatient treatments like Paxlovid may be game changing, but we have ruled out its effect because Paxlovid was approved by the European Medicines Agency only on January 28, 2022 and introduced in Spain very recently.

The implication of the present findings is that we obtained an early warning signal of ICU bed occupancy by COVID-19 patients that could be generated in the primary care setting and introduced in digital monitoring dashboards [29, 30]. The role of general practitioners in the health care system is therefore crucial for anticipating hospital pressure and for supporting policy decisions aimed at anticipating ICU saturation. We also note that applying our approach to other health systems requires consideration of the local context, criteria for hospital and ICU admission and reporting protocols. For example, in the US, there are 30 ICU beds/100,000 approximately [31, 32], which triples the rate in the region analyzed here.

In conclusion, the evolution of COVID-19-related ARI cases per 100,000 inhabitants is a simple tool for flagging up any future COVID-19 waves. Despite the complex situation and challenges posed by COVID-19 pandemic to health services, daily registry of COVID-19-related ARI diagnoses made in the primary care setting represents a novel and useful indicator of ICU bed occupancy in advance.

## Acknowledgments

The authors thank all the CatSalut personnel and all physicians involved in the project, the staff members of the SISAP and SISCAT for their cooperation in establishing and maintaining the database, and Marta Pulido, M.D., for editing the manuscript and editorial assistance.

## Author Contributions

**Conceptualization:** Montserrat Guillen, Ester Bordera Cabrera, Catalina Bolancé.

**Data curation:** Ignasi Bardes Robles, Ester Bordera Cabrera, Xénia Acebes Roldán.

**Formal analysis:** Montserrat Guillen, Catalina Bolancé, David Moriña.

**Methodology:** Montserrat Guillen, Catalina Bolancé, David Moriña.

**Resources:** Ignasi Bardes Robles, Ester Bordera Cabrera, Xénia Acebes Roldán.

**Software:** Montserrat Guillen, Catalina Bolancé.

**Validation:** Catalina Bolancé, David Moriña.

**Visualization:** Montserrat Guillen, Daniel Jorba.

**Writing – original draft:** Montserrat Guillen, Daniel Jorba.

**Writing – review & editing:** Montserrat Guillen, Ignasi Bardes Robles, Ester Bordera Cabrera, Xénia Acebes Roldán, Catalina Bolancé, Daniel Jorba, David Moriña.

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
