## [Decision Letter · Decision Letter 0]

28 Mar 2022

PONE-D-21-02244Acute Respiratory Infection Rates in Primary Care Anticipate ICU Bed Occupancy During COVID-19 WavesPLOS ONE

Dear Dr. Guillen,

Thank you for submitting your manuscript to PLOS ONE. After careful consideration, we feel that it has merit but does not fully meet PLOS ONE’s publication criteria as it currently stands. Therefore, we invite you to submit a revised version of the manuscript that addresses the points raised during the review process.

The present investigation is well designed and written. However, the main limitation is represented by the fact that data are somewha old (2020). We suggest the Authors to clearly explain why and how their findings are still clinically useful. 

We look forward to receiving your revised manuscript.

Kind regards,

Chiara Lazzeri

Academic Editor

PLOS ONE

Journal Requirements:

2. Thank you for stating the following financial disclosure: "This study was supported by grants from the Spanish Ministry of Science and Innovation (code PID2019-105986GB-C21) (MG, CB) and Instituto de Salud Carlos III (code COV20/00115) (DM), Madrid, Spain, and Institució Catalana de Recerca i Estudis Avançats  ICREA Academia (MG), Barcelona, Spain"

Reviewers' comments:

Reviewer's Responses to Questions

**Comments to the Author**

1. Is the manuscript technically sound, and do the data support the conclusions?

Reviewer #1: Yes

2. Has the statistical analysis been performed appropriately and rigorously? 

Reviewer #1: Yes

3. Have the authors made all data underlying the findings in their manuscript fully available?

Reviewer #1: Yes

4. Is the manuscript presented in an intelligible fashion and written in standard English?

Reviewer #1: Yes

5. Review Comments to the Author

Reviewer #1: This is an important question but it has also been informally understood since the beginning of the pandemic that outpatient cases � hospitalization 2 weeks later � death 4 weeks later (roughly). That said, they provide specificity for this general concept. Also, by tying it to GP visit they establish a somewhat uniform definition of outpatient severity (i.e. asymptomatic or very mild cases wouldn’t visit GP, generally).

I have some concerns about immediate relevance given that the data are in pandemic terms quite old (2020). On the other hand, the concept is an important one and I would therefore strongly support publication.

Introduction

A nice, concise description of what is known and the problem. Very well written.

Methods

Good description of patient population, appropriate to only use index visits for ARI with PCR confirmed COVID-19. Appropriate to include positive rapid antigen as specificity is high for those tests.

US has 27 ICU beds/100,000, so this should be addressed in the Discussion; applying this to other health systems will require consideration of the local context. Criteria for hospital and ICU admission may also vary. And, reporting requirements vary by country. This makes sense in a proper organized national health system, but not in places like my country which lack a true health system. Unfortunately.

Adjustment for holidays and weekends and day of week effects is important. Not being a biostatistician I am not able to judge whether your approach was appropriate, but it seem to be so.

Results

Again, very concise. Was the 17-day lag consistent throughout the study period? Also, this research was done with the ancestral variant. We are now on to omicron. Would be much more compelling if this study including much more recent data and looked at whether the lag was consistent despite changing variants and changing treatments. And, the availability of outpatient treatments like Paxlovid may be game changing.

6. PLOS authors have the option to publish the peer review history of their article (what does this mean?). If published, this will include your full peer review and any attached files.

Reviewer #1: **Yes: **Mark H. Ebell MD, MS

---

## [Author Response · Author response to Decision Letter 0]

8 Apr 2022

See file "Response to Reviewers.pdf"

---

## [Editor Report · Decision Letter 1]

11 Apr 2022

Acute Respiratory Infection Rates in Primary Care Anticipate ICU Bed Occupancy During COVID-19 Waves

PONE-D-21-02244R1

Dear Dr. Guillen,

We’re pleased to inform you that your manuscript has been judged scientifically suitable for publication and will be formally accepted for publication once it meets all outstanding technical requirements.

Kind regards,

Chiara Lazzeri

Academic Editor

PLOS ONE
---

## [Editor Report · Acceptance letter]

13 Apr 2022

PONE-D-21-02244R1 

Acute Respiratory Infection Rates in Primary Care Anticipate ICU Bed Occupancy During COVID-19 Waves 

Dear Dr. Guillen:

I'm pleased to inform you that your manuscript has been deemed suitable for publication in PLOS ONE. Congratulations! Your manuscript is now with our production department. 

Kind regards, 

on behalf of

Dr. Chiara Lazzeri 

Academic Editor

PLOS ONE